# Prevalence of Intestinal Parasites, Risk Factors and Zoonotic Aspects in Dog and Cat Populations from Goiás, Brazil

**DOI:** 10.3390/vetsci10080492

**Published:** 2023-07-31

**Authors:** Juliana Bruno Borges Souza, Zara Mariana de Assis Silva, Bruna Samara Alves-Ribeiro, Iago de Sá Moraes, Ana Vitória Alves-Sobrinho, Klaus Casaro Saturnino, Henrique Trevizoli Ferraz, Mônica Rodrigues Ferreira Machado, Ísis Assis Braga, Dirceu Guilherme de Souza Ramos

**Affiliations:** 1Laboratory of Parasitology and Veterinary Clinical Analysis, Institute of Agricultural Sciences, Federal University of Jataí, Jataí 75801-615, Brazil; julianabruno@discente.ufj.edu.br (J.B.B.S.); zaramariana@discente.ufj.edu.br (Z.M.d.A.S.); brunasamara@discente.ufj.edu.br (B.S.A.-R.); iago.moraes@discente.ufj.edu.br (I.d.S.M.); anasobrinho@discente.ufj.edu.br (A.V.A.-S.); isisbraga@ufj.edu.br (Í.A.B.); 2Laboratory of Veterinary Pathological Anatomy, Institute of Agricultural Sciences, Federal University of Jataí, Jataí 75801-615, Brazil; klaus.sat@ufj.edu.br; 3Laboratory of Veterinary Anatomy, Institute of Agricultural Sciences, Federal University of Jataí, Jataí 75801-615, Brazil; htferraz@ufj.edu.br; 4Laboratory of Biotechnology and Physiology in Fish, Institute of Biological Sciences, Federal University of Jataí, Jataí 75801-615, Brazil; monica_rodrigues@ufj.edu.br

**Keywords:** canine, feline, health, helminths, protozoa, zoonosis

## Abstract

**Simple Summary:**

Gastrointestinal diseases caused by parasites are frequently diagnosed in dogs and cats. This study aimed to identify the main intestinal parasites obtained from fecal samples of dogs and cats in the municipality of Jataí, Brazil, and associate them with risk factors. The eggs, cysts, and oocysts were identified as those of *Ancylostoma* spp., *Toxocara* spp., *Trichuris vulpis*, *Dipylidium caninum*, *Giardia* spp., *Entamoeba* spp., *Cystoisospora* spp., and *Platynosomum fastosum*. Considering the results, the need to implement preventive and control measures to reduce the occurrence of parasites in animals and the exposure of humans to pathogenic agents is evident.

**Abstract:**

Gastrointestinal diseases caused by parasites are frequently diagnosed in the clinical routine of domestic animals, especially dogs and cats. In general, they trigger factors that can affect human health due to zoonoses. Therefore, this study aims to identify the main intestinal parasites obtained from the fecal samples of dogs and cats in the municipality of Jata, Brazil, and their associated risk factors. Between October 2020 and March 2022, fecal samples were collected from 359 dogs and 55 cats through spontaneous defecation and subsequently subjected to coproparasitological analyses using the Willis fluctuation and Hoffman spontaneous sedimentation techniques. The following parasitic species were identified: *Ancylostoma* spp., *Toxocara* spp., *Trichuris vulpis*, *Dipylidium caninum; Giardia* spp., *Entamoeba* spp., *Cystoisospora* spp., and *Platynosomum fastosum*. The risk factors associated with parasitism include age, average income of owners, access to garbage, sewage, waste, outdated deworming, and contact with animals. The results demonstrate the need to establish public policies and implement preventive and control measures to reduce the occurrence of parasites in animals and the exposure of humans to pathogenic agents

## 1. Introduction

The proximity between humans and domestic animals, particularly dogs and cats, has increased considerably [1]. Pets act therapeutically in the lives of their guardians; however, their close coexistence facilitates the spread of zoonoses [1], which has been increasingly studied in the past ten years [1,2]. According to the World Health Organization (WHO) [3], approximately 24% of the world’s population is infected with soil-transmitted helminthiases, with the highest rates of occurrence being in sub-Saharan Africa, the Americas, China, and East Asia. Thus, it is necessary to take an interdisciplinary approach to reassess actions that can trigger the occurrence of parasites and have implications for the health of animals, humans, and the environment [4,5,6,7].

Gastrointestinal parasites have been frequently reported in domestic dogs and cats. The helminths and protozoa frequently diagnosed in these species include *Ancylostoma* spp., *Toxocara* spp., *Trichuris* spp., *Dipylidium caninum*, *Strongyloides stercoralis*, *Giardia* spp., *Cystoisospora* spp., and *Cryptosporidium* spp. [8,9,10,11,12,13,14,15]. Single infections or co-infections occur. In dogs, the prevalence of single infections ranged from 20.5% to 62.2% and co-infections from 16.1% to 37.4%, whereas in cats, they ranged from 46.2% to 90.9% and 3.3% to 41.4%, respectively [8,9,11,15,16,17].

The occurrence of clinical signs depends on the parasite species and parasite load, as well as the health status of the animal and the presence of co-infections, including in other systems (respiratory, etc.) [18]. Therefore, enteric infections can be asymptomatic [19] or progress with the onset of moderate-to-severe gastrointestinal disorders, development delay, and anorexia and may lead to patient death [20,21]. The clinical signs of parasitism by *Ancylostoma* spp. directly correlate with pathogenicity. Parasites of this genus undergo hematophagy in the host intestine, resulting in intense blood loss [22], and dysentery is frequently observed [23]. Parasitism by *Toxocara* spp. is associated with a prevalence of diarrhea, vomiting, abdominal distension, constipation, cough, and nasal secretion [24,25]. Entangled adults can lead to intestinal rupture and obstruction of the gallbladder and biliary and pancreatic ducts [26,27]. In humans, the parasites *Ancylostoma* spp. and *Toxocara* spp. are responsible for the occurrence of cutaneous larva *migrans* (CML) and visceral larva *migrans* (VLM), respectively. CML consists of serpiginous linear papular lesions with an inflammatory aspect [28]. In VLM, the clinical findings include hepatomegaly, eosinophilia, and multiple oval lesions in the liver and lungs [29].

Considering the importance of veterinary medicine in maintaining unique health and the recognition of the role of dogs and cats in the spread of parasitic zoonoses, this study aimed to identify the main gastrointestinal parasites affecting domestic animals by statistically evaluating the coproparasitological results and observing the risk factors associated with the occurrence of these parasites in dogs and cats in the municipality of Jataí, Brazil.

## 2. Materials and Methods

### 2.1. Location and Sampling

Between October 2020 and March 2022, fecal samples were collected from 359 dogs in the municipality of Jataí, Goiás, Brazil. The collections were carried out in the owners’ homes. The collections were distributed among 58 sectors and later separated into northern, western, eastern, central, and southern regions. All were carried out in the morning, with a single collection from each animal, through spontaneous defecation and sent to the Laboratory of Parasitology and Veterinary Clinical Analysis of the Federal University of Jataí. The sample calculation followed the design described by the following formula:$$n=\frac{N*Z_{\left(1-\alpha \right)}^{2}*[p*\left(1-p\right)]}{\left(N-1\right)*\;d^{2}+Z_{\left(1-\alpha \right)}^{2}*[p*\left(1-p\right)]}=\frac{33605*3.84*[0.5*\left(1-0.5\right)]}{\left(33604\right)*\;0.0025+3.84*[0.5*\left(1-0.5\right)]}=346$$
where *N* is the population size, which, according to Assis et al. (2020), was 33,605 dogs in Jataí in 2018; *Z*(1 − *α*) is the standard value of a normal distribution for a 95% confidence interval; *p* is an estimated proportion of 35% of parasitized animals (considering data from Martins et al. [30], who described an occurrence of parasitism of 34.41% in Mineiros, Goiás, adjacent municipality to Jataí); and *d* is required accuracy of 0.05 (5% maximum error).

### 2.2. Coproparasitological Examination

After collection, the stool samples were subjected to coproparasitological analysis using two techniques for researching helminth eggs and protozoan oocysts: Willis fluctuation [31] and spontaneous sedimentation by Hoffman et al. [32], adapted according to Hoffmann [33].

For flotation, a hypersaturated NaCl solution (NaCl 35%) was added to the stool samples and subsequently filtered through a sieve and gauze to remove debris, subjecting it to spontaneous fluctuation due to the difference in density of eggs and oocysts. Finally, they were observed between the slide and coverslip (with the addition of a drop of Lugol’s solution) using an optical microscope (Nikon Eclipse E200, Nikon, Tokyo, Japan).

The sedimentation test was carried out with the addition of water to the sample, filtered through a sieve and gauze to remove debris, subjected to spontaneous sedimentation in a specific cup, and then observed between the slide and coverslip (with the addition of a drop of Lugol) using optical microscopy (Nikon Eclipse E200). Eggs and oocysts were identified according to the method of Zajac and Conboy [34].

### 2.3. Data Analysis and Interpretation

The overall prevalence of each parasite species was calculated according to Bush et al. [35] by multiplying the number of positive animals relative to the total number of samples by 100 (%).

Along with the collection of each sample, an epidemiological questionnaire was used to compare parasitism with possible risk factors using the following variables: defined race; age; host sex; average income of owners; street access; presence of basic sanitation (sewage collection and piped water); access to sewage, garbage, and waste; updated deworming; and presence of contacting animals (including synanthropic animals). Factors associated with parasitism were analyzed using the Chi-Square (X^2^) test, considering a significant *p*-value < 0.05. Using the same test, the results of the two coproparasitological examinations were compared. The odds ratio (OR) was calculated to verify the level of risk associated with variables that correlated with parasitism.

Finally, we analyzed the spatial distribution of gastrointestinal parasites in dogs in the municipality of Jataí, Brazil.

### 2.4. Cats

Fecal samples were collected from 55 cats and subjected to the same coproparasitological analyses as previously described. As the “*n*” was small, the statistical analyses were not feasible, so only the occurrence descriptions were weighted in this study. The small number of samples is due to low receptivity to the study by cat owners.

### 2.5. Ethics Committee

The study was submitted for analysis by the Ethics in the Use of Animals Committee of the Federal University of Jataí for verification and monitoring of activities, having been approved by protocol 028/19.

## 3. Results

### 3.1. Prevalence and Richness of Parasites in Dogs

The number of observed species per sample ranged from zero to three (having hosts parasitized by one, two, or three species of parasites). Eggs of *Ancylostoma* spp., *Toxocara* spp., *Trichuris vulpis*, *D. caninum* egg capsules, cysts of *Giardia* spp. and *Entamoeba* spp., and oocysts of *Cystoisospora* spp. were identified (Figure 1).

The most prevalent parasites were *Ancylostoma* spp. (29.53%), followed by *Toxocara* spp. (7.52%), *D. caninum* (6.13%), *Cystoisospora* spp. (4.74%), *Giardia* spp. (3.34%), *T. vulpis* (1.67%), and *Entamoeba* spp. (0.84%). Data related to prevalence, absolute values, and confidence intervals are presented in Table 1. Co-infections were seen in 9.47% of animals and the most common was by *Ancylostoma* spp. and *Toxocara* spp. (3.9%). Six animals had a triple infection by *Ancylostoma* spp., *D. caninum*, and *Cystoisospora* spp. (*n* = 2); *Ancylostoma* spp., *Toxocara* spp., and *D. caninum* (*n* = 3); and *Ancylostoma* spp., *Toxocara* spp., and *Trichuris* spp. (*n* = 1).

### 3.2. Risk Factors

Risk factor analysis revealed that the following factors were associated with parasitism: age, average income of owners, access to garbage, sewage, waste, outdated deworming, and presence of contacting animals (including synanthropic animals). All *p*-values for the analyses are presented in Table 2.

The *odds ratio* of the variables associated with the risk of gastrointestinal parasitism revealed the following probabilities: adult animals are 116.7% more likely to be parasitized than elderly or young animals/puppies; animals whose guardians have a family income of up to BRL 3600.00 (approximately USD 705.88 in August 2022) are 81.5% more likely to be parasitized than animals whose guardians have an average income above this amount; animals with access to garbage, waste, and sewage are 17.1% more likely to be parasitized than animals that do not have contact with these elements; animals with outdated deworming are 644.4% more likely to be parasitized than animals with updated prophylaxis; and finally, animals that have contact with other animals, including synanthropic animals (rodents and pigeons), are 324.1% more likely to be parasitized than animals that are not in this condition.

The other analyzed variables showed no differences and, therefore, were not associated with parasitism.

### 3.3. Comparison of Techniques

In total, 95 samples were positive in both the sedimentation and fluctuation tests. Of the others, 31 samples were positive only for fluctuations and 27 were positive only for sedimentation. Despite this difference, when compared to the total number of parasites, the fluctuation technique had 82.89% sensitivity (126/152, confidence interval 76.91–88.88%) and the sedimentation technique had 79.61% sensitivity (121/152, confidence interval 72.65–85.59%), with no difference between them; thus, they complemented each other.

### 3.4. Spatial Distribution

The analysis of the spatial distribution showed that only the southern sector of the municipality of Jataí had a difference in relation to the others, which were considered statistically equal. The confidence intervals of prevalence intersect between the north (47.37%, confidence interval: 24.45–71.14), west (31.67%, confidence interval: 20.26–44.96), central (32.65%, confidence interval: 19.95–47.54), and east (31.78%, confidence interval: 23.87–40.56). The southern sector (65.69%, confidence interval 55.63–74.81), which despite having an intersection with the north does not have an intersection with the others, is considered an area of greater exposure (Figure 2).

### 3.5. Cats

Of the 55 animals sampled, 38 were positive for at least one gastrointestinal parasite species (69.09%). *Ancylostoma* spp. and *Toxocara* spp. eggs, *D. caninum* egg capsules, *P. fastosum* eggs, *Giardia* spp. cysts, and oocysts from *Cystoisospora* spp. were also found.

The most common parasites were *Ancylostoma* spp. (47.27%), followed by *Cystoisospora* spp. (29.09%), *D. caninum* (7.27%), *P. fastosum* (5.45%), *Toxocara* spp. (3.64%), and *Giardia* spp. (3.64%). Data related to frequencies, absolute values, and confidence intervals are presented in Table 3. Co-infections were seen in 27.27% of animals and the most common was by *Ancylostoma* spp. and *Cystoisospora* spp. (*n* = 13), followed by *Ancylostoma* spp. and *D. caninum* (*n* = 2).

## 4. Discussion

### 4.1. Prevalence of Parasites

According to Guimarães et al. [36], *Ancylostoma* is most commonly diagnosed in domestic animals in Brazil, mainly dogs. In the present study, this genus was the most prevalent parasite in dogs and cats, with a prevalence of 29.53% and 47.27%, respectively. *Ancylostoma* spp. is the genus that has high prevalence rates in all Brazilian regions, in addition to emphasizing the relevance of its zoonotic potential [37]. Other studies from the central-west region of Brazil found this species as the most prevalent in dogs and cats [30,38,39,40], evidencing the risk in regions of the Brazilian Cerrado (savannah) for humans (due to the zoonotic potential) and animals, including wild animals [41,42]. The high prevalence in Brazil, especially in the Cerrado biome, is related to environmental conditions, mainly in tropical areas, with temperatures between 25 and 30 °C. These factors favor the permanence and development of the parasite in the environment, making it a risk factor for infection in animals and humans. In addition, the non-restriction of animals unaccompanied by their guardians, and even stray animals, are factors described by Ribeiro et al. [43] and Prestes et al. [44] as contributors to the spread of parasites in the environment.

The genus *Toxocara* spp. is the second most frequently reported parasite in domestic and wild canids and felids. It is important to public health because of its transmission between animals and humans and is therefore characterized as a zoonosis. The prevalence in cats was 3.64%, which was within the expected range, and similar occurrences have been reported previously [45]. When released into the environment, eggs are resistant to environmental factors and are viable for long periods in soil. Thus, the environment in which infected animals are introduced and the environments they frequently encounter correspond to a high risk factor, resulting in damage to their unique health, to which humans and other animals are exposed daily. The ideal temperature for the eggs to develop varies from 15 to 30 °C, similar to that found in Jataí, (19.9 and 31.3 °C [46]), making it favorable for the development of eggs in the environment and resulting in a high prevalence of infections.

Parasitized animals often contaminate the environments in which they are located; therefore, humans are often exposed to parasitism, especially in places where temperature and humidity are favorable for the development of larvae, emphasizing housing as a possible risk factor. The frequency with which animals contaminate public places, such as squares and parks, is an important factor in the transmission of *Ancylostoma* spp. and *Toxocara* spp. [47], even more so in countries with a large number of animals with access to the street like in this study, as most of the animals evaluated have access to the street, including unaccompanied, free access to contaminated public places, increasing the probability of new infections and presenting risks to humans through exposure to pathogens, where infection can occur accidentally [48,49,50].

*Trichuris vulpis* is the most common parasite of the Trichuridae family reported in wild dogs and canids. Infection occurs through the oral route through the ingestion of water or soil contaminated with embryonated parasite eggs, but our prevalence is lower than in Cerrado areas with similar climates [40]. Eggs deposited in the environment take an average of 9–10 days to embryonate at higher temperatures; therefore, the environment in this municipality is favorable for them to develop and infect other animals [51]. A relevant fact is that Souza et al. [40] analyzed shelter animals that live in small and usually overcrowded environments that favor direct cycle parasites like *T. vulpis,* and in our study, we did not use shelter animals.

Cestodes and flukes are not the most common parasites in dogs and cats in the Brazilian Cerrado [30,38,39,40]. The prevalence of *D. caninum* was 6.13% and 7.27% for dogs and cats, respectively. Compared with other parasites in dogs and cats, *D. caninum* does not show a high prevalence [52,53]. Domestic and wild canids and felids are definitive hosts, in addition to having zoonotic potential and infecting humans. Infection occurs through the oral route in domestic and wild canids and felids, as well as in humans when the intermediate host was ingested (*Ctenocephalides* spp. and *Trichodectes canis*), containing the cysticercoid larva in their organism, and the presence of these intermediate hosts is more common in temperate than tropical zones [27,39]. Already *P. fastosum* has domestic and wild felids as definitive hosts, with great relevance in veterinary medicine, as it parasitizes the bile ducts and gallbladders of cats. Terrestrial mollusks of the genus *Subulina,* isopods, and lizards are intermediate hosts [54,55,56]. This study showed a prevalence of 5.45% in cats, which was consistent with most of the habits of the analyzed animals, which were mostly adults with a history of access to the street and hunting habits.

The low prevalence (3.34%) of *Giardia* spp. in dogs was because parasitized animals do not eliminate cysts continuously [57]. This protozoan has a wide variety of hosts and is frequently reported in dogs, cats, and humans. Infection occurs orally through ingestion of infective cysts in contaminated water, food, or soil. The habit of some owners not collecting the feces of their animals deposited in the environment is an eminent risk factor, favoring the contamination of soil and water and their occurrence in humans [58,59,60]. Unlike *Giardia* spp., *Cystoisospora* spp. is not considered a zoonotic agent [61]. Infection also occurs through the fecal–oral route and in places with precarious sanitary measures, frequently found in this study (access to sewage, garbage, and waste).

The genus *Entamoeba* has been reported in dogs, cats, cattle, horses, and humans. Transmission occurs orally after animals or humans ingest water or food contaminated with cysts [62,63]. The prevalence was 0.84% in dogs and was not diagnosed in cats, as such infections in cats are rare [64]. Despite the low occurrence in pets, we cannot disregard the possibility of transmission to humans (due to proximity) and that some species of *Entamoeba* genus are pathogenic [65,66].

### 4.2. Comparison of Techniques and Risk Factors

After collecting the stool samples, they were submitted to coproparasitological analysis, with the aim of looking for helminth eggs and protozoan oocysts. Two techniques were used, floating and spontaneous sedimentation. During the analyses, 95 samples tested positive in both the fluctuation and sedimentation tests. Of the others, 31 samples were positive only for fluctuations and 27 were positive only for sedimentation. Despite this difference, when compared to the total number of parasites, the fluctuation technique had 82.89% sensitivity (126/152), with a confidence interval of 76.91–88.88%. The sedimentation technique had a sensitivity of 79.61% (121/152) with a confidence interval of 72.65–85.59%. Therefore, there was no difference between them and they complemented each other, similar to the results of previous studies [67].

Adult dogs were more susceptible than elderly dogs or puppies in our studies. The prevalence of infection in adult dogs is approximately twice as high as that in puppies [68]. Magalhães et al. [13] in Minas Gerais, Brazil, demonstrated a similar prevalence in adult dogs; however, some studies have shown that young animals are more susceptible [69,70,71,72]. This divergence in prevalence is related to the parasite species and host characteristics. Younger animals are more susceptible to infections because of transplacental and transmammary transmission, and the immaturity of the immune system is also a contributing factor [69]. For example, *Ancylostoma* spp. demonstrate a higher occurrence in adult animals because of their longer exposure to pathogenic agents, especially in situations where they have an active infection in the environment in which they are found [8].

Family finances of up to BRL 3600.00 (approximately USD 728.97 in June 2023) increase the risk of parasitism in animals, which can be explained by the Brazilian economic profile. According to the Inter-Union Department of Statistics and Socioeconomic Studies (DIEESE) [73], the minimum wage required for a family of four is BRL 6298.91, which is equivalent to five nominal minimum wages. This estimate is based on basic needs such as food, housing, health, education, clothing, hygiene, transportation, leisure, and social security. Thus, for low-income owners, since their purchasing power is related to basic expenses, access to veterinarian support is less frequent or not at all. In addition, a high proportion of Brazilian families do not have basic sanitation conditions [74]. Nunes and Rocha [75] conducted a descriptive and cross-sectional study in Maceió, Brazil, where they concluded that most parasites in adolescents are due to a lack of basic sanitation and the presence of pet feces in the environment.

Contact with synanthropic animals, such as pigeons and rodents, increases the susceptibility to new parasitic infections by 324.1%. According to Paramasvaran et al. [76] and Allen et al. [77], the expansion of urban centers has contributed to the approximation of different animal species, including synanthropic species. This constant rapprochement between humans and domestic animals with synanthropy favors the transmission of various pathogenic agents, including gastrointestinal parasites. They can act as intermediate and paratenic hosts for pathogens and can spread contaminants in the environment, water, and food. It is worth noting the importance of implementing preventive control and prophylactic measures aimed at eradicating synanthropic animals to reduce the spread of pathogens [78].

In general, Jatai has an equivalent risk of exposure to pathogens, emphasizing the importance of one-health, which is a proportional risk for all, including animals and humans (especially due to the prevalence of zoonotic pathogens). Sanitary problems in municipalities, such as the intense presence of animals in the streets without the supervision of guardians and vacant lots with relatively high amounts of garbage, are found in different parts of the world, especially in developing countries. These data of Jataí are important for understanding and establishing public policies and implementing preventive actions [79]. Balassiano et al. [71] identified factors that intensify the occurrence of gastrointestinal parasites in dogs, including the lack of concern regarding the owners, who neglect these diseases; inefficient intercommunication between veterinarians and the population; lack of government programs aimed at controlling these conditions; high number of infected animals, contributing to new infections in animals exposed to risk factors; environmental contamination, through the non-removal of feces in public places during walks; and stray animals.

### 4.3. Control and Prevention (Special Comments)

Several factors contribute to the spread of pathogenic agents in the environment, including dogs and cats, which contaminate the environment through feces excreted in public places. Implementing health education for owners is fundamental to reducing the transmission of parasites. It is recommended that feces be collected during walks and immediately in domestic environments. Dogs and cats should not have access to the streets, and their guardians are responsible for the physical and environmental hygiene of their animals. In addition, access to garbage, waste, and sewage, which pose risks to them, should be restricted, as they may be contaminated and exposed [80,81]. Population control of stray animals is also necessary to control parasites, and owners should have access to veterinarians for guidance on the risks of zoonosis and health education [82,83,84].

Close contact among domestic animals, humans, and synanthropic animals is an important aspect to consider in the transmission of pathogenic agents. In the domestic environment, a balanced diet for animals with hygienic measures, garbage collection, and basic sanitation are important to control synanthropic animals and minimize the spread of pathogens, as evidenced in our study.

In kennels and catteries, it is essential to isolate contaminated areas, maintain continuous hygiene, use suitable products, and avoid the presence of animals. Additionally, sunlight during the day causes the environment to dry, preventing the development of eggs and larvae [85].

## 5. Conclusions

This study demonstrated a wide range of parasites in the coproparasitological analyses of dogs and cats in the municipality of Jataí, Goiás. In dogs, *Ancylostoma* spp., *Toxocara* spp., and *T. vulpis*; *D. caninum* egg capsules; cysts of *Giardia* spp. and *Entamoeba* spp.; and oocysts from *Cystoisospora* spp. were observed. In cats, eggs of *Ancylostoma* spp., *Toxocara* spp., *D. caninum* egg capsules, *P. fastosum* eggs, *Giardia* spp. cysts, and *Cystoisospora* spp. oocysts were observed. Risk factors related to parasitism included age, average income of owners, outdated deworming, presence of contacting and synanthropic animals, and access to garbage, sewage, and waste. Public policies to prevent and control measures and health education are vital for reducing the occurrence of parasites in animals and the exposure of humans to pathogenic agents.

## Figures and Tables

**Figure 1 vetsci-10-00492-f001:**
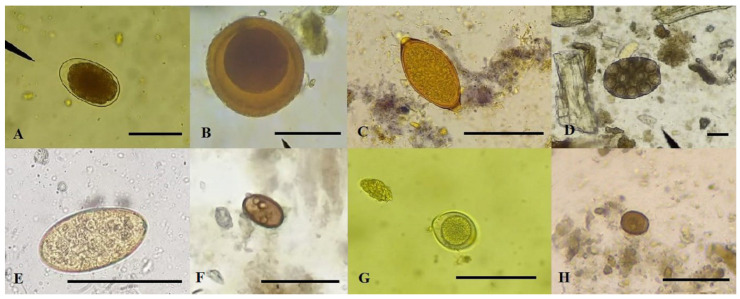
Eggs, cysts, and oocysts of intestinal parasites in fecal samples from dogs and cats in the municipality of Jataí, Brazil. (**A**) Egg of *Ancylostoma* spp. (bar: 50 µm). (**B**) Egg of *Toxocara* spp. (bar: 50 µm). (**C**) Egg of *Trichuris vulpis* (bar: 50 µm). (**D**) *Dipylidium caninum* ovigerous capsules (bar: 50 µm). (**E**) Egg of *Platynosomum fastosum* (bar: 50 µm). (**F**) Cyst of *Giardia* spp. (bar: 20 µm). (**G**) Oocyst of *Cystoisospora* spp. (bar: 40 µm). (**H**) Cyst of *Entamoeba* spp. (bar: 40 µm).

**Figure 2 vetsci-10-00492-f002:**
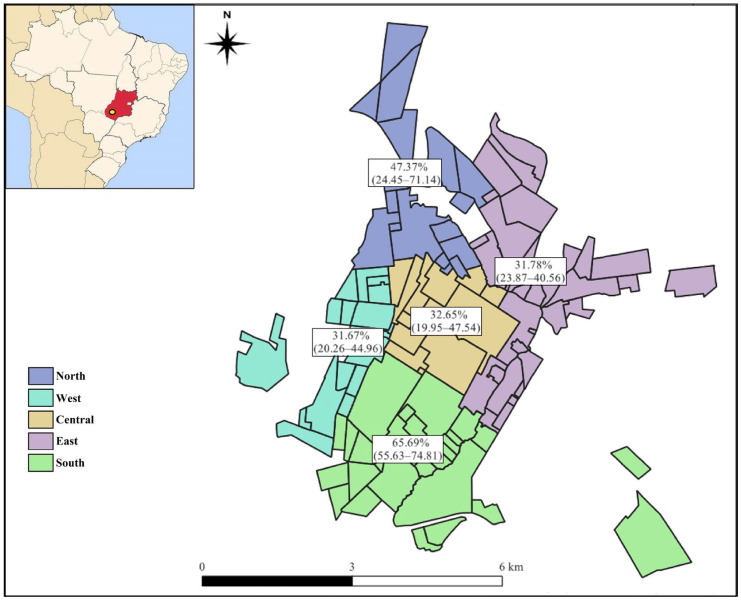
Map of the municipality of Jataí (highlighted in yellow in the upper left corner, location in Brazil), Goiás, with the prevalence and confidence intervals of the distribution of gastrointestinal parasites in dogs and cats.

**Table 1 vetsci-10-00492-t001:** Prevalence of gastrointestinal parasites in dog feces samples (*n* = 359) from Jataí, Brazil, submitted to coproparasitological examination between October 2020 and March 2022.

Species	Positives (*n*)	Prevalence (%)	Confidence Interval (%) *
Nematodes	116 **	32.31	27.68–37.31
*Ancylostoma* spp.	106	29.53	25.04–34.44 ^a^
*Toxocara* spp.	27	7.52	5.22–10.72 ^b^
*Trichuris vulpis*	6	1.67	0.77–3.60 ^c^
Cestodes	22 **	6.13	4.08–9.10
*Dipylidium caninum*	22	6.13	4.08–9.10 ^b^
Protozoa	32 **	8.91	6.38–12.31
*Giardia* spp.	12	3.34	1.92–5.75 ^bc^
*Cystoisospora* spp.	17	4.74	2.98–7.45 ^bc^
*Entamoeba* spp.	3	0.84	0.28–2.43 ^c^
Co-Infections	34	9.47	6.86–12.94

* Values with different letters differ at the 5% level of significance (comparisons only between the prevalence of parasite species). ** Group occurrence, including co-infections.

**Table 2 vetsci-10-00492-t002:** Associated risk factors and probability of occurrence of gastrointestinal parasites in fecal samples from dogs (*n* = 359) from Jataí, Brazil, submitted to coproparasitological examination between October 2020 and March 2022.

Risk Variable	*p*-Value *	*Odds Ratio ***
**Breed**	0.225	
Crossbreed		-
Breed		-
**Age**	0.005	
Juvenile/Pup		0.134
Adult		2.167
Elderly		0.246
**Host sex**	0.605	
Male		-
Female		-
**Average income of owner *****	0.007	
Up to BRL 3600.00		1.815
More than BRL 3600.00		0.551
**Street access**	0.318	
Yes		-
No		-
**Basic home sanitation (sewage collection and piped water)**	0.702	
Yes		-
No		-
**Access to sewage, garbage, and waste**	<0.001	
Yes		1.171
No		0.854
**Updated deworming**	<0.001	
Yes		0.134
No		7.444
**Presence of contacting animals (including synanthropic)**	0.019	
Yes		4.241
No		0.236

* Considered significant when *p*-value < 0.05. ** Calculated only for variables that obtained statistical significance. *** Reference value of approximately three minimum wages in Brazil (BRL) in the year 2023 (approximately USD 728.97, June 2023).

**Table 3 vetsci-10-00492-t003:** Frequency of gastrointestinal parasites in fecal samples from cats (*n* = 55) from Jataí, Goiás, Brazil, submitted to coproparasitological examination between October 2020 and March 2022.

Species	Positives (*n*)	Frequency (%)	Confidence Interval (%)
Nematodes	28	50.91	37.07–64.65
*Ancylostoma* spp.	26	47.27	33.65–61.20
*Toxocara* spp.	2	3.64	0.44–12.53
Cestodes	4	7.27	2.02–17.59
*Dipylidium caninum*	4	7.27	2.02–17.59
Flukes	3	5.45	1.14–15.12
*Platynosomum fastosum*	3	5.45	1.14–15.12
Protozoa	18	32.73	20.68–46.71
*Giardia* spp.	2	3.64	0.44–12.53
*Cystoisospora* spp.	16	29.09	17.23–42.90
Co-Infections	15	27.27	16.14–40.6

## Data Availability

All datasets are available in the main manuscript. The dataset supporting the conclusions of this article is included within the article.

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
