# Peer review of "Prevalence of Intestinal Parasites, Risk Factors and Zoonotic Aspects in Dog and Cat Populations from Goiás, Brazil"

_vetsci, 2023, doi:10.3390/vetsci10080492_

Round 1

Reviewer 1 Report

This is an interesting paper which is part of continuous monitoring of parasites in dogs and cats in different areas of the world- this paper focuses on Brazil. There is a large number of canid samples used, but a smaller number of felid samples, which is a shame, but felid samples are often harder to obtain so this is understandable. It is nicely analysed throughout, with some good points raised

Major points

My biggest point with this paper is the discussion. You have some interesting results, but many have been released previously. Your discussion focusses very heavily on the clinical signs of the parasites which is not really a big part of the study It is also too long, and can be cut down by about a third if not more. You have bits which repeat the introduction, and other bits which repeat the results. I suggest rewriting the discussion to bring into context the results, and how the prevalence values are similar to the other countries in the region and how that varies. How do the risk factors which you report match those from other studies?

Minor comments

Line 28 and 35- what do you mean by clinical routine?

Line 42-43- you need to include spp., here in relevant places

Line 64-65- this seems unclear- single and co infections usually occur with more than one etiological agent? How can a single infection involve more than one agent? Please reword

Line 76-77- the bit about death in hatchlings seems a bit strange and unclear- please reword or remove

Line 126- How was parasite richness investigated?

Lines 150-152 and 159-161- please ensure that parasite species names which are Latin are in italics

Line 174- please put significant in full here

A few minor comments are detailed above

Author Response

Reviewer 1

This is an interesting paper which is part of continuous monitoring of parasites in dogs and cats in different areas of the world- this paper focuses on Brazil. There is a large number of canid samples used, but a smaller number of felid samples, which is a shame, but felid samples are often harder to obtain so this is understandable. It is nicely analysed throughout, with some good points raised

Answer: Thank you very much for your willingness to read our manuscript and help us to improve it. We wanted a larger sample of feline feces, however, cat owners were not as receptive to the study as dog owners. We describe this in lines 143-144.

Major points

My biggest point with this paper is the discussion. You have some interesting results, but many have been released previously. Your discussion focusses very heavily on the clinical signs of the parasites which is not really a big part of the study It is also too long, and can be cut down by about a third if not more. You have bits which repeat the introduction, and other bits which repeat the results. I suggest rewriting the discussion to bring into context the results, and how the prevalence values are similar to the other countries in the region and how that varies. How do the risk factors which you report match those from other studies?

Answer: The entire discussion was reviewed and rewritten as suggested by reviewers 1 and 2.

Minor comments

Line 28 and 35- what do you mean by clinical routine?

Answer: We tried to specify that it was frequently diagnosed by Veterinarians. However, we noticed the redundancy and changed it to: “Gastrointestinal diseases caused by parasites are frequently diagnosed in dogs and cats”.

Line 42-43- you need to include spp., here in relevant places.

Answer: Changed as per suggestion.

Line 64-65- this seems unclear- single and co infections usually occur with more than one etiological agent? How can a single infection involve more than one agent? Please reword

Answer: Rewritten as suggested.

Line 76-77- the bit about death in hatchlings seems a bit strange and unclear- please reword or remove

Answer: We removed the phrase.

Line 126- How was parasite richness investigated?

Answer: We removed the term, as in fact there was no richness analysis.

Lines 150-152 and 159-161- please ensure that parasite species names which are Latin are in italics

Answer: Checked as per suggestion.

Line 174- please put significant in full here.

Answer: Changed as per suggestion.

Reviewer 2 Report

OVERALL COMMENTS

This study aimed to investigate the prevalence of intestinal parasites in the canine and feline populations living in the municipality of Jata, Goiás, Brazil, and the associated risk factors. The authors highlight the importance of parasite control in companion animals from a One Health perspective, analyzing also social and economic aspects as risk factors for the spread of zoonoses within the community. The manuscript is well written and the results are well showed. However, the discussion section needs to be revised and condensed in some of its parts. In addition, few more data regarding co-infections can maybe be better provided. Therefore, in my opinion, the manuscript is suitable for publication in Veterinary Sciences, pending minor revisions.

MINOR COMMENTS

Title

I woud recommend addressing the manuscript’s title with the name of the place where the study has been conducted. For instance, the title could be “Prevalence of intestinal parasites and their associated risk factors in dog and cat populations from Goiás, Brazil”.

Abstract

Line 37: Please replace “individual health” with “human health”.

Introduction

Lines 51 – 53: These two sentences sounds more like general assumptions of the Authors. Although the meaning is understandable, these phrases need to be supported by some references.

Line 54: “Increasingly studied” – Please indicate a time period to refer to for this expression. For example, in the last 5/10/20 years…

Lines 53 – 56: Condense these two sentences in one single phrase.

Line 56: Replace “important” with “necessary”.

Lines 57 – 60: Move this sentence above the previous one as follows to help the reader in understanding the passage from the human to the animal aspect/approach: “ …it is an important issue as it affects animal, human and environmental health. According to the World Health Organization (WHO, 2022) [7], approximately 24% of the world's population is infected with soil-transmitted helminthiases, with the highest rates of occurrence being in sub-Saharan Africa, the Americas, China, and East Asia. Thus, it is necessary to take an interdisciplinary approach to reassess actions that can trigger the occurrence of parasites [3-6].”

Lines 64 – 68: Are these data obtained in other prevalence studies in Brasil? Please specify where the prevalence data have been obtained and in which host type (i.e., owned dogs/sheltered dogs or stray dogs).

Line 69: This is true. However, clinical signs depend also on the general health status of the animal, and the presence of co-infections with other endoparasites not just intestinal ones. I would suggest specifying it in the text.

Line 71: Later on, in the discussion, the authors use the expression “development delay” instead of “growth retardation”. I would suggest adopting the first one (i.e., development delay) and being consistent throughout the whole text.

Lines 84 – 86: This phrase is a bit redundant as already discussed at the beginning of the paragraph.

M&M

Line 98: Please write “from each animal” instead of “of each animal”.

Results

Lines 150 – 162: Please check the italics.

Discussions

Lines 243 – 247: This is redundant as already partially explained in the introduction. I would suggest delete these two sentences while integrating the missing concepts in the introduction.

Line 249: “canids” instead of “canines”.

Lines 269 – 277: This paragraph seems more part of an introduction than of a discussion about Trichuris sp. prevalence. Please rephrase it accordingly.

Line 280: “felids” instead of “felines”.

Line 288: Remove the “P” from “PPlatynosum”.

Table 1

Could you also provide the percentage of infection for each group (i.e., nematodes, cestodes and Protozoa)? And provide also co-infections prevalence? Same for Table 3.

Table 2

For “race”, do you mean “breed”? If so, I suggest using the terms “breed” and “crossbreed”. If info on dog’s breed are available, could you assess the risk factors correlated to a specific breed and discuss those results? (for instance, different breed can have different behaviour and this might change the probability of exposure to parasitic infections).

Author Response

Reviewer 2

OVERALL COMMENTS

This study aimed to investigate the prevalence of intestinal parasites in the canine and feline populations living in the municipality of Jata, Goiás, Brazil, and the associated risk factors. The authors highlight the importance of parasite control in companion animals from a One Health perspective, analyzing also social and economic aspects as risk factors for the spread of zoonoses within the community. The manuscript is well written and the results are well showed. However, the discussion section needs to be revised and condensed in some of its parts. In addition, few more data regarding co-infections can maybe be better provided. Therefore, in my opinion, the manuscript is suitable for publication in Veterinary Sciences, pending minor revisions.

Answer: Thank you very much for your willingness to read our manuscript and help us to improve it. We considered all suggestions and modified the discussion.

MINOR COMMENTS

Title

I woud recommend addressing the manuscript’s title with the name of the place where the study has been conducted. For instance, the title could be “Prevalence of intestinal parasites and their associated risk factors in dog and cat populations from Goiás, Brazil”.

Answer: Changed as per suggestion.

Abstract

Line 37: Please replace “individual health” with “human health”.

Answer: Changed as per suggestion.

Introduction

Lines 51 – 53: These two sentences sounds more like general assumptions of the Authors. Although the meaning is understandable, these phrases need to be supported by some references.

Answer: The sentences are supported by reference [1]. We added citation.

Line 54: “Increasingly studied” – Please indicate a time period to refer to for this expression. For example, in the last 5/10/20 years…

Answer: Changed as per suggestion.

Lines 53 – 56: Condense these two sentences in one single phrase.

Answer: Changed as per suggestion.

Line 56: Replace “important” with “necessary”.

Answer: Changed as per suggestion.

Lines 57 – 60: Move this sentence above the previous one as follows to help the reader in understanding the passage from the human to the animal aspect/approach: “ …it is an important issue as it affects animal, human and environmental health. According to the World Health Organization (WHO, 2022) [7], approximately 24% of the world's population is infected with soil-transmitted helminthiases, with the highest rates of occurrence being in sub-Saharan Africa, the Americas, China, and East Asia. Thus, it is necessary to take an interdisciplinary approach to reassess actions that can trigger the occurrence of parasites [3-6].”

Answer: Changed as per suggestion.

Lines 64 – 68: Are these data obtained in other prevalence studies in Brasil? Please specify where the prevalence data have been obtained and in which host type (i.e., owned dogs/sheltered dogs or stray dogs).

Answer: We agree with the suggestion, however, in the introduction we prefer to keep the information, so we brought this approach in the restructuring of the discussion.

Line 69: This is true. However, clinical signs depend also on the general health status of the animal, and the presence of co-infections with other endoparasites not just intestinal ones. I would suggest specifying it in the text.

Answer: Changed as per suggestion.

Line 71: Later on, in the discussion, the authors use the expression “development delay” instead of “growth retardation”. I would suggest adopting the first one (i.e., development delay) and being consistent throughout the whole text.

Answer: Changed as per suggestion.

Lines 84 – 86: This phrase is a bit redundant as already discussed at the beginning of the paragraph.

Answer: We changed to “In VLM the clinical findings include hepatomegaly, eosinophilia, and multiple oval lesions in the liver and lungs [29]”. We removed the after sentence as well.

M&M

Line 98: Please write “from each animal” instead of “of each animal”.

Answer: Changed as per suggestion.

Results

Lines 150 – 162: Please check the italics.

Answer: Checked and changed as per suggestion.

Discussions

Lines 243 – 247: This is redundant as already partially explained in the introduction. I would suggest delete these two sentences while integrating the missing concepts in the introduction.

Answer: The entire discussion was reviewed and rewritten as suggested by reviewers 1 and 2.

Line 249: “canids” instead of “canines”.

Answer: Changed as per suggestion.

Lines 269 – 277: This paragraph seems more part of an introduction than of a discussion about Trichuris sp. prevalence. Please rephrase it accordingly.

Answer: The entire discussion was reviewed and rewritten as suggested by reviewers 1 and 2

Line 280: “felids” instead of “felines”.

Answer: Changed as per suggestion.

Line 288: Remove the “P” from “PPlatynosum”.

Answer: Changed as per suggestion.

Table 1

Could you also provide the percentage of infection for each group (i.e., nematodes, cestodes and Protozoa)? And provide also co-infections prevalence? Same for Table 3.

Answer: Added as per suggestion.

Table 2

For “race”, do you mean “breed”? If so, I suggest using the terms “breed” and “crossbreed”. If info on dog’s breed are available, could you assess the risk factors correlated to a specific breed and discuss those results? (for instance, different breed can have different behaviour and this might change the probability of exposure to parasitic infections).

Answer: We changed to Breed as suggested. However, under the breed information, the problem is that the number of crossbreed animals is much (much higher) than the breed animals.

Reviewer 3 Report

Please write latin names of parasites in italics

Better to replace Ancylostoma with hookworms, why not Uncinaria?

Some details regarding Dipylidium proglotids

Author Response

Reviewer 3

Please write latin names of parasites in italics

Answer: Thank you very much for your willingness to read our manuscript and help us to improve it. We changed as per suggestion

Better to replace Ancylostoma with hookworms, why not Uncinaria?

Answer: We consider placing as Ancylostomatidae. However, in several parasitological necropsies carried out in the region in the last eight years, we did not find Unicnaria in domestic animals.

Some details regarding Dipylidium proglotids

Answer:  We did not find D. caninum proglottids, but ovigerous capsules.

Round 2

Reviewer 1 Report

The authors have done a good job with improving the manuscript and it now reads much better. There are a few final very minor points which I have detailed below

Line 54- therefore it is a necessary issue …. (reword)

Line 159- Co-infections were seen in 9.47% of animals the most common …. (reword)

Table 2- Up to BRL 3,600.00 may sound better here

Line 254- Co-infections were seen in 27.27% of animals the most common …. (reword)

Line 291- can delete frequently in here

Line 334- Entamoeba needs to be in italics

Line 367- and cross sectional study in Maceio, Brazil …. (reword)

Line 407- a balanced diet for animals with hygienic measures …. (reword)

It was a pleasure to be asked to review the manuscript, and I wish the authors all the best for their future research

A few very minor comments above

Author Response

The authors have done a good job with improving the manuscript and it now reads much better. There are a few final very minor points which I have detailed below.

Answer: We really appreciate every contribution. Other corrections follow.

Line 54- therefore it is a necessary issue …. (reword)

Answer: We move to lines 58-60 for better connection “Thus, it is necessary to take an interdisciplinary approach to reassess actions that can trigger the occurrence of parasites and due o implications for the health of animals, humans, and the environment.”

Line 159- Co-infections were seen in 9.47% of animals the most common …. (reword)

Answer: We changed as suggested.

Table 2- Up to BRL 3,600.00 may sound better here

Answer: We changed as suggested.

Line 254- Co-infections were seen in 27.27% of animals the most common …. (reword)

Answer: We changed as suggested.

Line 291- can delete frequently in here

Answer: We changed as suggested.

Line 334- Entamoeba needs to be in italics

Answer: We changed as suggested.

Line 367- and cross sectional study in Maceio, Brazil …. (reword)

Answer: We changed as suggested.

Line 407- a balanced diet for animals with hygienic measures …. (reword)

Answer: We changed as suggested.

It was a pleasure to be asked to review the manuscript, and I wish the authors all the best for their future research

Answer: Thank you very much. We hope the same for you.